# Plasma Metabolomics to Evaluate Progression of Necrotising Enterocolitis in Preterm Pigs

**DOI:** 10.3390/metabo11050283

**Published:** 2021-04-29

**Authors:** Yan-Nan Jiang, Yong-Xin Ye, Per Torp Sangild, Thomas Thymann, Søren Balling Engelsen, Bekzod Khakimov, Ping-Ping Jiang

**Affiliations:** 1School of Public Health, Sun Yat-sen University, Guangzhou 510080, China; jiangyn25@mail2.sysu.edu.cn (Y.-N.J.); yeyx9@mail2.sysu.edu.cn (Y.-X.Y.); 2Section for Comparative Paediatrics and Nutrition, Department of Veterinary and Animal Sciences, University of Copenhagen, DK-1870 Frederiksberg, Denmark; pts@sund.ku.dk (P.T.S.); thomas.thymann@sund.ku.dk (T.T.); 3Department of Neonatology, Rigshospitalet, DK-2100 Copenhagen, Denmark; 4Department of Paediatrics, Odense University Hospital, DK-5000 Odense, Denmark; 5Department of Food Science, University of Copenhagen, DK-1958 Frederiksberg, Denmark; se@food.ku.dk (S.B.E.); bzo@food.ku.dk (B.K.)

**Keywords:** necrotising enterocolitis (NEC), antibiotics, metabolomics, amino acids, lipid metabolism

## Abstract

Necrotising enterocolitis (NEC) is a microbiome-dependent gut disease in preterm infants in early life. Antibiotic treatment is a common intervention for NEC. How NEC lesions, with or without antibiotics, affect plasma metabolome was explored in this study. Formula-fed preterm pigs were used as a model for human NEC and treated with saline, parenteral or oral antibiotics (*n* = 15–17) for four days after delivery. Gut tissues were collected for evaluation of NEC-like lesions and plasma for metabolomic analysis by proton nuclear magnetic resonance spectroscopy (^1^H-NMR). Metabolites were annotated, quantified and subjected to statistical modelling to delineate the effects of NEC and antibiotic treatment. Presence of severe NEC lesions, not antibiotic treatment, was the main drive for plasma metabolite changes. Relative to other pigs, pigs with severe NEC lesions had higher levels of alanine, histidine and *myo*-inositol, and lower levels of 3-hydroxybutyric acid and isobutyric acid. Across NEC lesion states (healthy, mild, severe), antibiotics directly affected only a few metabolites (tryptophan, 3-phenyllactic acid). Together and independently, NEC and antibiotic treatment affected circulating metabolites in preterm pigs. Amino acids and plasma metabolites, partly related to the gut microbiome, may be helpful to monitor progression of NEC lesions after proper validation.

## 1. Introduction

Necrotising enterocolitis (NEC) is a serious gut inflammatory disease affecting 3-10% of hospitalised preterm infants and has a high mortality and many co-morbidities [1]. Risk factors for NEC include immature gut, excessive gut bacterial colonization and aggressive feeding, especially when using formula [2,3]. Modulations of bacterial colonization of the gut with pre-, pro- or antibiotics have therefore been tested as preventive or therapeutic interventions for NEC. Whilst reports on human infants linked antibiotics to increased incidence of NEC and neonatal sepsis [4,5], other earlier studies documented NEC-preventive effects of oral [6,7] and systemic [8] antibiotic treatment. Concerns for increased antibiotic resistance also limit the use of antibiotics for NEC prevention [9], and the most effective regimen of antibiotic treatment against NEC remains elusive.

Both antibiotic treatment and NEC progression can alter the composition of the gut microbiome and its metabolism [10,11], in turn affecting the host metabolism and circulating metabolites in preterm infants [12], including amino acid and lipid derivatives [13,14]. Differences were found in plasma and urinary metabolites between control and antibiotics-treated (NEC-protected) preterm pigs, although effects specific to antibiotic treatment or NEC were difficult to determine [15]. In early life metabolites partly derived from the gut, such as amino acids and short-chain fatty acids (SCFAs), may influence the development of distant organs, such as the brain, or be important for overall body growth [16,17]. If such plasma metabolites show changes in the early phase of NEC, they may serve as biomarkers for prediction and early diagnosis of NEC [13,18]. Explorative studies on plasma metabolic changes in response to NEC progression in infants are limited by the relatively rare and variable clinical conditions of such infants and by the limited blood volume that can be sampled from such vulnerable infants. In addition, widely differing antibiotic regimens, potentially having effects on metabolites independent of the NEC effects, make the interpretation of results difficult. Animal models of preterm birth, allowing different modes of antibiotic treatment, with and without spontaneous NEC development, are required for discovering such metabolites. Metabolites affected only when NEC lesions become severe may help in the decision-making about continued medical treatment (antibiotics, enteral food withdrawal) or surgical intervention.

In this study a preterm pig model of NEC was used to investigate the plasma metabolites associated with NEC development and two different antibiotic protocols, oral or systemic antibiotic treatment for four days after birth. It has been previously reported in detail how such treatments, especially oral antibiotic treatment, reduced NEC sensitivity and improved gut structure/function, together with delayed gut bacterial colonisation and SCFA production [19], as well as the effects on systemic immunity [20] and plasma proteins [21]. We hypothesized that NEC development, independently or together with antibiotic treatment, would affect plasma metabolites. The plasma metabolome was profiled by untargeted metabolomics based on one-dimensional proton nuclear magnetic resonance spectroscopy (1D ^1^H-NMR), and metabolites were annotated and quantified using an in-house software developed to process complex NMR metabolomics spectra, Signature Mapping (SigMa) [22].

## 2. Results

In this study, formula-fed preterm pigs were used as a model for infant NEC and treated with saline (CON, *n =* 15), parenteral (PAR, *n =* 17) or oral antibiotics (ORA, *n =* 15). After four days of antibiotic treatment, gut tissue was collected for NEC evaluation with the in-house scoring system at euthanisation, and plasma was collected for metabolomic analysis.

### 2.1. Clinical Observations and NEC Lesions

As listed in Appendix A, body weights (BW) were similar for the CON, PAR and ORA pigs at birth (mean ± SEM: 894 ± 56, 916 ± 44 and 916 ± 56 g, respectively), but over the next four days, the CON pigs grew slower than the PAR and ORA pigs (11.3 ± 2.4, 19.0 ± 1.3 and 16.2 ± 2.6 g/kg/d, respectively). Remaining values for clinical data, blood biochemistry and haematology, and organ weights are available in our previous publication [19].

Overall, a lower incidence of NEC (NEC score ≥ 3 in any gut region) was observed in the ORA pigs (0 out of 15), relative to the CON pigs (9 out of 15, *p* = 0.001, Fisher’s exact test) or the PAR pigs (10 out of 17, *p* = 0.001, Fisher’s exact test). Distribution of the NEC severity in the CON, PAR and ORA groups is shown in Appendix A, and the NEC scores (median, interquartile range Q1 and 3) of the CON, PAR and ORA pigs were 4 (1.5–5), 3 (1–4) and 1 (1–2), respectively. Among the pigs with NEC (NEC score ≥ 3 in at least one gut region), the PAR pigs tended to have more cases of Mild-NEC (score 3–4) than the CON pigs (7 out of 10 vs. 2 out of 9, *p* = 0.07, Fisher’s exact test), and the incidences of NEC lesions in different gut regions (stomach, prox, mid and distal small intestine or colon) were not different (*p* = 0.21, Fisher’s exact test), as summarized in Appendix A.

### 2.2. Plasma Metabolites

A representative graph of NMR spectra acquired is shown in Figure 1. A PCA score plot of all detected NMR features is shown in Appendix A. No clear effect of NEC or antibiotic treatment on the overall metabolome was observed. A total of 27 metabolites were annotated. Information of all annotated metabolites, including name, molecular formula, chemical shift, abundance in different groups and effect size, is summarised in Appendix A. Nine and five metabolites showing significant difference in abundance among the NEC groups and antibiotic groups are listed in Table 1 and Table 2, respectively.

No metabolites differed between the Mild-NEC and No-NEC (score 1–2) groups, whilst seven differed between the Severe-NEC (score 5–6) and No-NEC pigs (*p* < 0.05 and |effect size| ≥ 0.8). Five metabolites differed between the Severe-NEC and Mild-NEC groups. The observed metabolites with differential abundance are involved in physiological aspects including metabolisms of amino acids, fatty acids, carbohydrates and energy. For plasma cholesterol, levels detected by NMR correlated well with previous data from bioassays (Pearson’s *r* = 0.89, *p* < 0.05) [19]. Levels of tyrosine, 3-isobutyric acid and cholesterol were lower and *myo*-inositol higher in the Severe-NEC relative to No-NEC pigs with large effect size (> 1.00 in absolute value). Levels of histidine and formic acid were significantly higher in the Severe-NEC, compared with Mild-NEC pigs (*p* < 0.05, |effect size| > 1.00). Tryptophan, 3-phenyllactic acid and ethanol were only affected by the antibiotic treatment, not by NEC. Levels of tryptophan were significantly higher in the ORA pigs than in the CON pigs. Levels of 3-phenyllactic acid increased from the CON and PAR to the ORA pigs. The PAR pigs had higher levels of ethanol, significantly different from the other two groups (both *p* < 0.05).

## 3. Discussion

Based on our statistical modelling, the presence of NEC lesions on day 5 of life, rather than the antibiotic treatment during the previous days, was the key driver of the metabolomic changes in plasma revealed by NMR. These effects on metabolites were mainly found in the pigs with severe NEC lesions, indicating that plasma metabolite changes did not appear until the NEC lesions were relatively advanced, at least according to our NEC scoring system. These scores may or may not closely associate with the clinical scoring systems for infants, such as Bell’s NEC stages. Regardless, the metabolites associated with clearly visible NEC lesions at autopsy reflect how NEC influences the metabolism of amino acids, fatty acids, carbohydrate and energy. Several of these metabolite changes may have occurred via changes to the gut microbiome in these pigs, following NEC development or the antibiotic treatment [19], which remains to be investigated in more detail. The metabolite changes did not result from differential growth rates, as no close correlation was found between the metabolite changes and growth rates of the pigs (all Spearman’s *r* < 0.5, data not shown).

The levels of three amino acids in plasma, namely alanine, histidine and tyrosine, were associated with NEC, as shown by the statistical modelling. Given the identical nutritional regimen for all pigs, changes of these amino acids can be attributed to the NEC itself. The increased levels of alanine found in this study were in line with a previous report on the urinary metabolome of NEC human patients [23], but unlike in that study, the alanine: histidine ratio was not changed in our study. Similar to findings from infants [14], tyrosine levels were the lowest in the pigs with severe NEC, whereas the levels of phenylalanine, its precursor, were unaffected. Increased phenylalanine levels and phenylalanine: tyrosine ratio have been observed in NEC infants and linked to NEC-induced limitation in hepatic conversion of phenylalanine to tyrosine, related to oxidative stress [13]. In this manner, other blood amino acids are also dynamically affected by both nutrition (parenteral vs. enteral nutrition) and endogenous metabolism, potentially making longitudinal patterns of multiple amino acids more sensitive in reflecting NEC-related metabolic changes, rather than amino acid levels at a single time-point.

The levels of two short-chain fatty acids (SCFAs), formic acid and isobutyric acid, and a fatty acid metabolite, 3-hydroxybutyric acid, were associated with NEC lesions. A significant correlation was also found between the levels of isobutyric acid and valine (*r* = 0.72), suggesting that the change in isobutyric acid levels is attributed to valine, the precursor of isobutyric acid, as previously reported [24]. In a quail model of NEC high caecal levels of isobutyric acid correlated with NEC, potentially via increased activity of Clostridia [25]. Consistent with this, Clostridia was the dominating genus in the CON pigs that suffered most NEC, as previously reported [19].

The levels of 3-hydroxybutyric acid, a ketone body, were lower in severe NEC pigs, relative to other pigs, and correlated with the levels of its precursor, acetoacetic acid (Spearman’s *r* = 0.49). Ketone bodies are generated by β-oxidation of free fatty acids, including SCFAs, in hepatocytes and enterocytes [26]. Reduced ketogenesis is potentially related to increased systemic inflammation in the NEC pigs, as indicated by our earlier studies [20], reducing the hepatic uptake of free fatty acids to generate 3-hydroxybutyric acid [27]. In addition, local NEC lesions, especially in the colon region, might have affected the colonic conversion of butyric acid into 3-hydroxybutyric acid.

Disturbed carbohydrate metabolism has been repeatedly observed in the gut tissues of NEC-sensitive pigs [28], and the resultant changes in plasma metabolites may be highly time-dependent during the disease progression. In this study the levels of central glycolytic metabolites (*myo*-inositol, pyruvate) were elevated, together with reduced glucose levels, in pigs with severe NEC lesions. Increased plasma levels of *myo*-inositol were also observed in septic pigs [29] and infants [30], and are consistent with the hypothesis that newborns with low energy stores rely on aerobic glycolysis to mount immune responses [31]. In human adults an increase in *myo*-inositol levels has been linked to insulin resistance and hyperglycaemia [32]. Hyperglycaemia is common in infants with NEC [33], at least in its early phase, and has been associated with an increase in late mortality [34], without notable effects of insulin on NEC or other morbidities [35]. The tendency to lower glucose levels in pigs with severe NEC indicates that glucose stores are depleted and gluconeogenesis fails to restore its levels.

Pyruvate, an intermediate of energy metabolism, was found as a marker of tissue hypoxia [36], a process associated with NEC [37]. In our previous study tissue expression of *HIF1A* increased in pigs with NEC [38]. Hypoxia-inducible factor-1α (HIF-1α) mediates the adaptation to hypoxia by suppressing the tricarboxylic acid cycle (TCA), resulting in pyruvate accumulation [39]. Pyruvate is then metabolised into lactate and alanine [39]. Correspondingly, pyruvate levels in the preterm pigs in this study were highly correlated with lactate (Spearman’s *r* = 0.90) and alanine levels (*r* = 0.83).

As revealed by statistical modelling, the PAR and ORA groups had different metabolites affected, relative to the controls, potentially resulting from their differential effects on the gut microbiome [19]. Conversely, the higher tryptophan levels in the ORA pigs are supported by a previous study [40] and may involve the AB suppression of tryptophan-catabolising bacteria, such as the *Clostridium* and *Bacteroides* genera [41]. Likewise, higher levels of 3-phenyllactic acid (PLA), a bacterial catabolite of phenylalanine [42], in AB-treated pigs may result from a higher activity of the phenylalanine-catabolising bacterial genera, including Lactobacilli and Staphylococci [43,44], as previously indicated in preterm pigs [19]. Higher plasma PLA levels were also found in patients with septic shock [45], who may have a distorted gut microbiome. On the other hand, PLA levels did not increase in a previous study on AB treatment of preterm pigs [15], indicating that microbiota-dependent plasma metabolite levels may depend on the type, dose and treatment regimen of antibiotics, in turn affecting the gut microbiome composition and activity.

## 4. Materials and Methods

### 4.1. Animal Procedure, NEC and Antibiotic Treatment

Delivery, rearing, feeding of preterm pigs and their antibiotic treatment were carried out as previously described [19] upon ethical approval (the Danish National Animal Experimentation Board, Copenhagen, Denmark, No. 2014-15-0201-00418). Briefly, pigs were delivered by caesarean section on day 106 of gestation (90% gestation) from three sows (Large White × Danish Landrace × Duroc). Immediately after birth, the pigs were weighed and fitted with umbilical arterial catheters and orogastric feeding tubes and immunised with maternal plasma. Based on birth weight (BW) and sex, a total of 47 pigs were block-randomized into three groups: pigs receiving parenteral (PAR, n = 17) or oral (ORA, n = 15) antibiotics or receiving saline orally (CON, n = 15) as controls. Ampicillin (30 mg/kg birth weight, three times daily), gentamicin (2.5 mg/kg BW, two times daily) and metronidazole (10 mg/kg BW, three times daily), formulated for parenteral or oral use, were adopted as previously reported [15] and listed in Appendix A.

Parenteral nutrition (PN) was provided as 4 mL/kg/h in the first 24 h, gradually increasing to 6–8 mL/kg/h. Minimal enteral nutrition (MEN) was given as a bolus of 3 mL/kg BW every 3 h, initiating within 5 h after delivery. PN plus MEN in the first two days transited to total enteral nutrition (15 mL/kg every 3 h) on day 3, and this was kept until the euthanasia on day 5. The nutritional compositions of PN and the formula used are listed in Appendix A.

On day 5, intracardial blood samples were collected from pigs under anaesthesia. EDTA-treated plasma samples were separated and stored at −80 °C for further analysis. The collected gastrointestinal tracts were divided into five regions (stomach, proximal, middle and distal small intestine, and colon) for macroscopic scoring of NEC lesions using our previously validated scoring system [19], where score 1 represents absence of macroscopic haemorrhage, oedema or mucosal abnormality; score 2 represents local hyperaemia; score 3 represents hyperaemia, extensive oedema and local haemorrhage; score 4 represents extensive haemorrhage; score 5 represents local necrosis and pneumatosis intestinalis; and score 6 represents extensive transmural necrosis and pneumatosis intestinalis. Representative pictures of the gut tissues can be seen in our previous publication [21]. The maximal NEC score across the five regions was used to classify the pigs into No-NEC (NEC score 1–2, *n =* 28), Mild-NEC (NEC score 3–4, *n =* 9) and Severe-NEC (NEC score 5–6, *n =* 10) groups for statistical analysis.

### 4.2. H-NMR Based Metabolomics

All 47 plasma samples and 4 extra pooled plasma samples, generated by pooling aliquots of all samples, were subjected to ^1^H-NMR analysis, as previously described [46]. Thawed plasma samples were mixed with phosphate buffer solution of the same volume (pH 7.4, 50 mM) [47], then transferred into SampleJet tubes (103.5 mm × 5 mm). NMR spectra were recorded on a Bruker Avance III 600 MHz NMR spectrometer (Bruker Biospin, Rheinstetten, Germany) equipped with a 5 mm broadband inverse RT (BBI) probe, automated tuning and matching accessory (ATMA), a cooling unit BCU-05 and an automated sampler (SampleJet, Bruker Biospin, Rheinstetten, Germany). Spectra were acquired using a standard pulse sequence with water suppression (Bruker pulse program library *noesygppr1d*). Thirty-two free induction decays (FIDs) were collected into 98,304 data points in a spectral width of 30 ppm. Prior to Fourier transformation, FIDs were subjected to apodization using an exponential function corresponding to a 0.3 Hz line-broadening. Spectra were automatically phase- and baseline-corrected using TOPSPIN 3.5 PL6 (Bruker BioSpin, Rheinstetten, Germany). The NMR spectra were processed using our in-house software, Signature Mapping (SigMa) [22], which facilitates direct identification and quantification of metabolites. Abundances of signature signals of metabolites were used to calculate absolute concentrations using the Electronic REference To access In vivo Concentrations (ERETIC) method, as described previously [48].

### 4.3. Statistical Analysis

The data matrix, containing metabolite identities and concentrations, was combined with other grouping information, such as litter (sow identity), NEC severity and sex, and exported into R (version 3.6.3, R Core Team, Vienna, Austria) [49], interfaced with R Studio [50], for subsequent analysis. Abundance of a metabolite was fitted to a linear mixed-effect model using the package nlme [51], with the NEC severity and antibiotic treatment groups as the fixed-effect factors. Litter was included as a random-effect factor. The factor of variance inflation (vif) was calculated to test possible variance inflation of each model fitted. Any model with a vif < 2 was regarded as acceptable. Pair-wise comparison was further conducted to test the difference between the NEC groups (No-NEC, Mild-NEC and Severe-NEC) or antibiotic treatment groups (CON, PAR and ORA) using the package multcomp [52]. With regard to the explorative nature of this study, leaving the lowering of the type I error a priority, the *p* value adjustment was not conducted, as advocated against by Feise [53]. Instead, the effect size (the magnitude of effect), calculated using the equation below [54], was included as a selection criterium [53], based on Cohen’s suggestion [55]. Any metabolite with *P* value < 0.05 between any two levels of either NEC severity or antibiotic treatment and an effect size (in absolute value) ≥ 0.8 was included for a functional assignment below. Principal component analysis (PCA), Fisher’s exact test and Spearman’s correlation were also performed in R.
(1)$$Effect\;size=\frac{Difference\;between\;the\;least\;squared\;means}{\sqrt{var_{intercept}+var_{slope}+var_{residual}}}$$

## 5. Conclusions

In this study, the clinical conditions with antibiotic treatments to preterm infants during (suspected/confirmed) NEC were mimicked in our pig model, and a range of both NEC- and antibiotic-related metabolite changes were found, aided by statistical modelling to separate the effects of NEC and the antibiotic treatment. Metabolites involved in the metabolism of fatty acids, such as isobutyric acid and 3-hydroxybutyric acid, are of potential as putative biomarkers for NEC, as they are involved in NEC pathogenesis, such as ischemia. The plasma metabolites affected by severe NEC found in preterm pigs may or may not reflect the metabolites affected in preterm infants during the later phase of NEC progression. It is not possible to directly translate the spontaneous development of NEC in preterm pigs into the conditions in infants. Nevertheless, the results provide a proof of concept for the independent and interacting effects of NEC progression and antibiotic treatment on the levels of plasma metabolites. Specifically, most plasma metabolites are not affected until NEC lesions become relatively severe, potentially reflecting a need for surgical intervention. More studies are required to verify our findings in human infants to better understand the roles of such plasma metabolites in the early and later phases of NEC in preterm infants.

## Figures and Tables

**Figure 1 metabolites-11-00283-f001:**
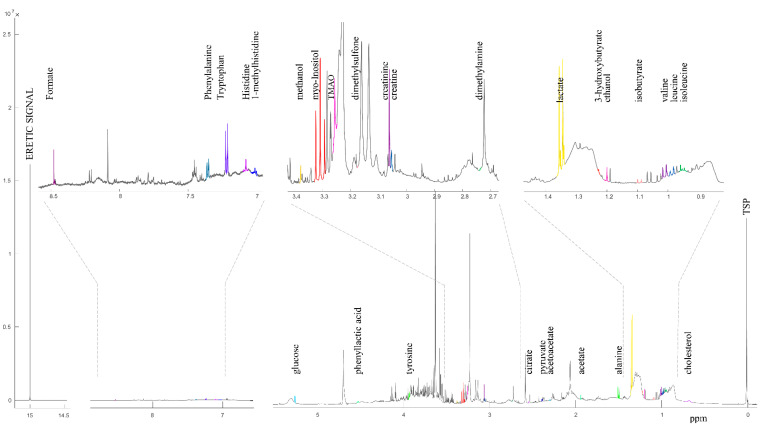
A representative one-dimensional proton (^1^H) NMR spectrum of pig blood plasma sample and SigMa-identified signature signals of selected blood metabolites. Intensity is scaled to the maximum of 2.5 × 10^7^.

**Table 1 metabolites-11-00283-t001:** Metabolites with differential abundance associated with NEC severity.

Metabolite	Molecular Formula	Chemical Shift (δ, ppm)	Multiplicity	Grouping	Abundance by NEC Severity (Mean ± SEM, mM)			Effect Size ^1^		
					No-NEC (*n* = 28)	Mild-NEC (*n* = 9)	Severe-NEC (*n* = 10)	Mild- NEC vs. No-NEC	Severe-NEC vs. No-NEC	Severe-NEC vs. Mild-NEC
Alanine	C_3_H_7_NO_2_	1.51	d	Amino acid	0.92 ± 0.05	0.93 ± 0.09	1.40 ± 0.42	−0.16	0.85 ^#^	1.01 ^#^
Histidine	C_6_H_9_N_3_O_2_	7.08	s	Amino acid	0.16 ± 0.01	0.15 ± 0.01	0.21 ± 0.04	−0.35	0.75	1.10 *
Tyrosine ^2^	C_9_H_11_NO_3_	3.93	dd	Amino acid	3.84 ± 0.17	3.84 ± 0.35	2.73 ± 0.45	0.15	−1.07 *	−1.22 *
Pyruvate	C_3_H_4_O_3_	2.39	s	Energy metabolism	0.13 ± 0.01	0.13 ± 0.02	0.20 ± 0.04	−0.12	0.86 ^#^	0.99 ^#^
Creatine ^2^	C_4_H_9_N_3_O_2_	3.05	s	Energy metabolism	0.24 ± 0.03	0.21 ± 0.02	0.15 ± 0.03	0.02	−0.84 ^#^	−0.85
3-Hydroxybutyric acid	C_4_H_8_O_3_	1.23	d	Ketone	0.05 ± 0.01	0.04 ± 0.01	0.01 ± 0.01	−0.76	−1.11 *	−0.35
Formic acid ^2^	HCOOH	8.48	s	SCFA	0.13 ± 0.04	0.10 ± 0.01	0.35 ± 0.14	−0.37	0.73	1.10 *
Isobutyric acid ^2^	C_4_H_7_O_2_H	1.10	d	SCFA	0.10 ± 0.00	0.10 ± 0.01	0.08 ± 0.01	−0.17	−1.10 *	−0.93 ^#^
Glucose ^2^	C₆H₁₂O₆	5.26	d	Carbohydrate metabolism	4.64 ± 0.24	4.39 ± 0.52	3.67 ± 0.55	−0.15	−0.97 *	−0.82
Cholesterol	C_27_H_46_O	0.68	m	Lipid metabolism	3.91 ± 0.13	4.00 ± 0.36	3.04 ± 0.44	0.09	−1.01 *	−1.10 *
*myo*-Inositol	C_6_H_12_O_6_	3.31	d	Carbohydrate metabolism	8.37 ± 0.72	8.93 ± 1.05	14.40 ± 1.93	0.12	1.24 **	1.11 **
Methanol	CH_3_OH	3.38	s	Carbohydrate metabolism	0.10 ± 0.01	0.12 ± 0.01	0.14 ± 0.01	0.64	1.02 *	0.38

SCFA, short-chain fatty acid. ^1^ Effect size was calculated for pairwise comparisons in the mixed effects analysis; Tukey test was used to calculate *p* value for pairwise comparison. ** *p* < 0.01; * *p* < 0.05; ^#^ *p* < 0.1. ^2^ Log2-transformed data were used.

**Table 2 metabolites-11-00283-t002:** Metabolites with differential abundance associated with antibiotic treatment.

Metabolite	Molecular Formula	Chemical Shift (δ, ppm)	Multiplicity	Grouping	Abundance by Antibiotic Treatment (Mean ± SEM, mM)			Effect Size ^1^		
					CON (*n =* 15)	PAR (*n =* 17)	ORA (*n =* 15)	PAR vs. CON	ORA vs. CON	ORA vs. PAR
Tryptophan	C_11_H_12_N_2_O_2_	7.21	m	Amino acid	0.29 ± 0.03	0.41 ± 0.07	0.53 ± 0.09	0.46	0.99 *	0.54
Phenylalanine	C_9_H_11_NO_2_	7.35	m	Amino acid	0.19 ± 0.02	0.24 ± 0.02	0.22 ± 0.01	0.80 ^#^	0.64	−0.16
3-Phenyllactic acid	C_9_H_10_O_3_	4.53	dd	Amino acid derivative	0.38 ± 0.04	1.32 ± 0.03	1.70 ± 0.06	1.66 **	2.33 **	0.68 **
3-Hydroxybutyric acid	C_4_H_8_O_3_	1.23	d	Ketone	0.04 ± 0.01	0.05 ± 0.02	0.03 ± 0.01	0.21	−0.68	−0.89 *
Formic acid ^2^	HCOOH	8.48	s	SCFA	0.26 ± 0.11	0.18 ± 0.04	0.07 ± 0.01	0.44	−0.50	−0.94 *
Ethanol	C_2_H_5_OH	1.20	t	Carbohydrate metabolism	0.45 ± 0.05	0.57 ± 0.03	0.46 ± 0.03	0.76 *	−0.23	−0.99 **
Citrate	C₆H₈O₇	2.55	d	Carbohydrate metabolism	0.41 ± 0.04	0.54 ± 0.06	0.46 ± 0.03	0.71 ^#^	−0.15	−0.86 ^#^

SCFA, short-chain fatty acid. ^1^ Effect size was calculated for pairwise comparisons in the mixed effects analysis; Tukey test was used to calculate *p* value for pairwise comparison. ** *p* < 0.01; * *p* < 0.05; ^#^ *p* < 0.1. ^2^ Log2-transformed data were used.

## Data Availability

The datasets analysed during the current study are available from the corresponding author on reasonable request.

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
