# Peer review of "Plasma Metabolomics to Evaluate Progression of Necrotising Enterocolitis in Preterm Pigs"

_metabolites, 2021, doi:10.3390/metabo11050283_

Round 1
Reviewer 1 Report
This is a high fidelity model for preterm NEC and well written and designed exploratory study. It also has some potential clinical relevance given the widespread use of systemic antibiotics in the NICU.
Minor comments only.
Page 2, line 85 is there an error in the NEC scoring here as the denominator is both 15 and 17?
Page 3, line 110 I would not use 'tendencies' as you have small subject numbers but large data outcomes with multiple tests and so you are at risk of finding significant associations that are meaningless, please only report significant findings, i.e. focus on the 7 different.
The discussion is appreciated as brief and balanced, not over interpreting findings that as you point our are likely secondary to NEC in process and not predictive of NEC or causative. Agree the microbiome work is important next step.
Reviewer 2 Report
In the manuscript by Jiang et al, the authors evaluate blood metabolites of formula fed piglets as a model of NEC with and without concomitant antibiotics treatment. Authors conclude that pigs with severe NEC lesions had significant differences in some metabolites from those with milder disease or no NEC. Furthermore, few differences were observed between the various antibiotic treatments. Finally, they suggest that plasm metabolites can serve as biomarkers for NEC.
Although this is a very interesting concept and NEC biomarkers would be very clinically relevant, there are a number of significant limitations to this analysis and the manuscript as a whole.
- The introduction is biased to suggest that initial treatment of premature infants with antibiotics reduces the rates of NEC. While this might be true for the specific publication sited, there is a lot of controversy in this area that was not included in the manuscript. There is actually substantial literature that antibiotic treatment increases the rates of NEC. Although data suggest that gut microbiota do play a role in the pathogenesis of NEC, how, when and which antibiotics are administered to reduce the rates of NEC are unknown. This should be addressed in the manuscript.
- The data on NEC histology and regions affected are not included in the manuscript and only referenced per previous publication. At the minimum, the numbers of animals with NEC should be included in a table in the manuscript with the regions affected in each group. Similarly, the authors conclude that the rates of NEC were highest in the control group. I’m not convinced that 9/15 (control group) is much higher than 10/17 (par group). Although the distribution of NEC severity is shown in fig S-1, the numbers should be included in a table as well.
- Statistical considerations need to be addressed. The authors state that “Due to the limited number of annotated metabolites, P value adjustment was deemed unnecessary and not conducted.” I am not sure this is valid. There were 27 metabolites detected and as such adjusted p value should be reported.
- It is unclear how the metabolite differences in the NEC samples and the antibiotic treatments were computed. For the analysis of the antibiotics on the blood metabolites, were all the patients with NEC excluded? If not, wouldn’t this skew the data?
- The suggestion that plasma metabolites can be used as a biomarker in NEC in the abstract and main text is purely speculative and should be excluded.
- There are some mislabeling of tables/figures referenced in the paper. For example line 78 should say table S-3 not S-1.
Reviewer 3 Report
See attached file

Round 2
Reviewer 2 Report
all of my concerns have been addressed